# Comparing Oncological and Perioperative Outcomes of Open versus Laparoscopic versus Robotic Radical Nephroureterectomy for the Treatment of Upper Tract Urothelial Carcinoma: A Multicenter, Multinational, Propensity Score-Matched Analysis

**DOI:** 10.3390/cancers15051409

**Published:** 2023-02-23

**Authors:** Nico C. Grossmann, Francesco Soria, Tristan Juvet, Aaron M. Potretzke, Hooman Djaladat, Alireza Ghoreifi, Eiji Kikuchi, Andrea Mari, Zine-Eddine Khene, Kazutoshi Fujita, Jay D. Raman, Alberto Breda, Matteo Fontana, John P. Sfakianos, John L. Pfail, Ekaterina Laukhtina, Pawel Rajwa, Maximillian Pallauf, Cédric Poyet, Giovanni E. Cacciamani, Thomas van Doeveren, Joost L. Boormans, Alessandro Antonelli, Marcus Jamil, Firas Abdollah, Guillaume Ploussard, Axel Heidenreich, Enno Storz, Siamak Daneshmand, Stephen A. Boorjian, Morgan Rouprêt, Michael Rink, Shahrokh F. Shariat, Benjamin Pradere

**Affiliations:** 1Department of Urology, Comprehensive Cancer Center, Medical University of Vienna, 1090 Vienna, Austria; 2Department of Urology, University Hospital Zurich, 8091 Zurich, Switzerland; 3Department of Urology, Luzerner Kantonsspital, 6004 Lucerne, Switzerland; 4Division of Urology, Department of Surgical Sciences, San Giovanni Battista Hospital, University of Studies of Torino, 10124 Turin, Italy; 5Department of Urology, Lions Gate Hospital, North Vancouver, BC V7L 2L7, Canada; 6Department of Urology, Mayo Clinic, Rochester, MN 55902, USA; 7Department of Urology, USC/Norris Comprehensive Cancer Center, University of Southern California, Los Angeles, CA 90033, USA; 8Department of Urology, St. Marianna University School of Medicine, Kawasaki 214-8525, Japan; 9Department of Urology, University of Florence, Careggi Hospital, 50100 Florence, Italy; 10Department of Urology, University of Rennes, 35000 Rennes, France; 11Department of Urology, Kindai University Faculty of Medicine, Osaka 589-8511, Japan; 12Department of Urology, Penn State Health, Hershey, PA 17033, USA; 13Urology Department, Fundació Puigvert, Autonomous University of Barcelona, 08025 Barcelona, Spain; 14Department of Urology, Icahn School of Medicine at Mount Sinai, New York, NY 10029, USA; 15Institute for Urology and Reproductive Health, Sechenov University, 119991 Moscow, Russia; 16Department of Urology, Medical University of Silesia, 40-752 Katowice, Poland; 17Department of Urology, Paracelsus Medical University Salzburg, University Hospital Salzburg, 5020 Salzburg, Austria; 18Department of Urology, The James Buchanan Brady Urological Institute, The Johns Hopkins University School of Medicine, Baltimore, MD 21231, USA; 19USC Institute of Urology, Keck Medicine of USC, University of Southern California, Los Angeles, CA 90007, USA; 20Department of Urology, Erasmus MC Cancer Institute, University Medical Center, 3015 GD Rotterdam, The Netherlands; 21Department of Urology, Azienda Ospedaliera Universitaria Integrata of Verona, University of Verona, 37126 Verona, Italy; 22Vattikuti Urology Institute, Henry Ford Hospital, Detroit, MI 48202, USA; 23Department of Urology, UROSUD, La Croix Du Sud Hospital, 31130 Quint-Fonsegrives, France; 24Department of Urology, Uro-Oncology, Robot Assisted and Specialized Urologic Surgery, University Hospital Cologne, 50937 Cologne, Germany; 25Urology Department, GRC n°5, Predictive Onco-Uro, AP-HP, Pitié-Salpêtrière Hospital, Sorbonne University, 75013 Paris, France; 26Department of Urology, University Medical Center Hamburg-Eppendorf, 20251 Hamburg, Germany; 27Hourani Center for Applied Scientific Research, Al-Ahliyya Amman University, Amman 19328, Jordan; 28Karl Landsteiner Institute of Urology and Andrology, 1090 Vienna, Austria; 29Department of Urology, Weill Cornell Medical College, New York, NY 10065, USA; 30Department of Urology, University of Texas Southwestern, Dallas, TX 75390, USA; 31Department of Urology, Second Faculty of Medicine, Charles University, 150 06 Prague, Czech Republic

**Keywords:** RNU, UTUC, transitional cell carcinoma, treatment outcomes, surgical approach

## Abstract

**Simple Summary:**

The growth of minimally invasive techniques for radical nephroureterectomy (RNU) has significantly changed the surgical treatment landscape of non-metastatic upper urinary tract urothelial carcinoma in recent decades. The aim of this study was to compare perioperative and oncologic outcomes between open, laparoscopic, and robotic RNU using a retrospective, multicenter, multinational database. Using 756 propensity-score-matched patients out of a total of 2434, we found a worse bladder recurrence-free survival in patients undergoing laparoscopic and robotic RNU compared with open RNU. Recurrence-free, cancer-specific, and overall survival were similar between the three surgical approaches. Laparoscopic and robotic RNU revealed a shorter hospital length of stay and fewer major postoperative complications compared to open RNU. Although minimally invasive RNU techniques are associated with improved perioperative outcomes, further studies are warranted to investigate the underlying factors responsible for the worse bladder recurrence-free survival of patients treated with these techniques.

**Abstract:**

Objectives: To identify correlates of survival and perioperative outcomes of upper tract urothelial carcinoma (UTUC) patients undergoing open (ORNU), laparoscopic (LRNU), and robotic (RRNU) radical nephroureterectomy (RNU). Methods: We conducted a retrospective, multicenter study that included non-metastatic UTUC patients who underwent RNU between 1990–2020. Multiple imputation by chained equations was used to impute missing data. Patients were divided into three groups based on their surgical treatment and were adjusted by 1:1:1 propensity score matching (PSM). Survival outcomes per group were estimated for recurrence-free survival (RFS), bladder recurrence-free survival (BRFS), cancer-specific survival (CSS), and overall survival (OS). Perioperative outcomes: Intraoperative blood loss, hospital length of stay (LOS), and overall (OPC) and major postoperative complications (MPCs; defined as Clavien–Dindo > 3) were assessed between groups. Results: Of the 2434 patients included, 756 remained after PSM with 252 in each group. The three groups had similar baseline clinicopathological characteristics. The median follow-up was 32 months. Kaplan–Meier and log-rank tests demonstrated similar RFS, CSS, and OS between groups. BRFS was found to be superior with ORNU. Using multivariable regression analyses, LRNU and RRNU were independently associated with worse BRFS (HR 1.66, 95% CI 1.22–2.28, *p* = 0.001 and HR 1.73, 95%CI 1.22–2.47, *p* = 0.002, respectively). LRNU and RRNU were associated with a significantly shorter LOS (beta −1.1, 95% CI −2.2–0.02, *p* = 0.047 and beta −6.1, 95% CI −7.2–5.0, *p* < 0.001, respectively) and fewer MPCs (OR 0.5, 95% CI 0.31–0.79, *p* = 0.003 and OR 0.27, 95% CI 0.16–0.46, *p* < 0.001, respectively). Conclusions: In this large international cohort, we demonstrated similar RFS, CSS, and OS among ORNU, LRNU, and RRNU. However, LRNU and RRNU were associated with significantly worse BRFS, but a shorter LOS and fewer MPCs.

## 1. Introduction

Urothelial cancers are the sixth most common tumors in developed countries, of which upper tract urothelial cancers (UTUC) account for only 5–10% [1]. Indeed, UTUC is a rare disease with a worse prognosis when compared to similarly staged bladder cancer [2]. Treatment of high-risk, non-metastatic UTUC is mainly based on radical nephroureterectomy (RNU) with bladder cuff excision (BCE), with or without lymph node dissection (LND) [3]. The growth of minimally invasive techniques for RNU has significantly changed the surgical treatment landscape of UTUC. The use of laparoscopic (LRNU) and robotic (RRNU) RNU increased from 36% to 54% between 2004 and 2013 [4]. However, survival and perioperative outcomes of LRNU and RRNU (compared to ORNU) have been mostly assessed in small-cohort studies [5,6,7,8,9,10] or from national databases [4,11,12,13,14,15,16,17] with associated weaknesses in design. Therefore, the current evidence on the survival and perioperative outcomes remains weak, and the available studies show large heterogeneity in statistical approaches. Thus, the present work aimed to provide more robust data on the three different surgical approaches (ORNU, LRNU, and RRNU) and present their impact on survival and perioperative outcomes in patients with high-risk non-metastatic UTUC.

## 2. Materials and Methods

### 2.1. Cohort Description and Patient Management

This was a retrospective, multicenter study involving 21 academic centers from Europe, Asia, and the United States. Patients’ medical records were retrospectively screened to identify individuals with high-risk UTUC treated with open, laparoscopic, or robotic RNU with BCE between 1990 and 2020. Exclusion criteria were clinical metastatic disease (cM+ or cN+) at the time of surgery, history of prior cystectomy due to bladder cancer, and missing follow-up. A predefined data set was used to collect patient information from the medical records of each center, and data were anonymized prior to sharing. The study was approved by the local ethics committees of all participating institutions (protocol code 1566/2017).

All RNU procedures were performed using standard techniques [18,19,20] with BCE in all included patients. The decision to perform open, laparoscopic, or robot-assisted RNU, as well as a lymphadenectomy and its extent, was at the surgeon’s discretion and was performed according to the standard templates described previously [21]. Pathologic examinations of the surgical specimens were performed by genitourinary pathologists at each participating center. Tumor stage was evaluated using the 2002 Union for International Cancer Control’s Tumor, Node, Metastasis Classification System. Tumor grade was assessed by the World Health Organization classification of 2004/2016. For cases before 2002 and 2004, a pathological review with restaging and regrading was performed. 

### 2.2. Follow-Up and Outcome Measurement

The primary objective of this study was to assess the impact of the surgical approach on survival outcomes, specifically with recurrence-free survival (RFS), bladder recurrence-free survival (BRFS), cancer-specific survival (CSS), and overall survival (OS) as the primary endpoints. Recurrence was defined as any local or distant recurrence/metastasis after RNU, and RFS was defined as the time interval between RNU and the first detection of recurrence. Bladder recurrence was considered separately, and BRFS was defined as the time interval between the date of RNU and first detection of bladder recurrence at follow-up cystoscopy. CSS was defined as the time interval between the date of RNU and disease-related death. OS was defined as the time interval between the time of RNU and death of any cause. The schedule for follow-up was based on international recommendations at the time of treatment and on the physicians’ preference. In general, surveillance included cross-sectional imaging and cystoscopy every three months for the first two years, semiannually from the second to the fifth year, and then annually. Cause of death was determined by the treating physicians and confirmed by a chart review and/or death certificates [22].

The secondary objective was to evaluate the association of the surgical approach with perioperative characteristics. Secondary endpoints included intraoperative blood loss, hospital length of stay, overall complications within 30 days postoperative (Clavien–Dindo classification 1–5), and major postoperative complications (MPCs) within 30 days postoperative (Clavien–Dindo classification ≥ 3). 

### 2.3. Statistical Analysis

To assess the differences between categorical variables, Pearson’s chi-squared or Fisher’s exact test were used, and for continuous or ordinal variables, the Kruskal–Wallis rank-sum test was used. 

Missing data (Appendix A) were assumed to be missing at random and were replaced with a multiple imputation of chained equations [23] by using 15 data sets to minimize the simulation error (Monte Carlo) [24]. For the imputation of numerical data, predictive mean matching was used, and for binary data, logistic regression imputation was used. For categorical data, polytomous logistic regressions (factor data with unordered levels >2) or classification and regression trees (factor data with ordered levels >2) were performed.

Strip plots were generated depicting the discrepancy between the observed and imputed data for verifying the plausibility of imputations (Appendix A) [24]. Next, a 1:1:1 propensity score matching (PSM) analysis was performed to reduce selection bias and adjust for the effects of unbalanced covariates. Adjusting variables were chosen which influence survival outcomes but do not relate to the exposure (surgical procedure) in order to increase the precision of the estimated exposure effect without increasing bias [25] (adjusting variables are listed in Appendix A). The balance of covariates before and after propensity score matching was assessed by visual inspection using love plots to indicate mean differences (Appendix A). 

Kaplan–Meier curves were used to estimate survival outcomes, and pairwise log-rank tests were utilized to compare these between the three different surgical approaches. Univariable and multivariable Cox regression analyses were performed to examine the association between surgical approaches and survival outcomes. Univariable and multivariable linear and logistic regression analyses were performed to examine the association between surgical approaches and perioperative outcomes. In the multivariable analysis, all variables were included, which demonstrated a statistically significant association with the corresponding dependent variable in the univariable analysis. Statistical significance was set at *p* < 0.05. All tests were two-sided. All statistical analyses were performed using R (version 4.0.3, Vienna, Austria, 2020).

## 3. Results

### 3.1. Study Cohort and Clinicopathologic Characteristics 

A total of 2434 patients met our inclusion criteria. The median percentage of missing data for the variables was 11% (IQR 6.25, 23.1) (Appendix A). After PSM, we obtained a cohort of 756 patients, with 252 patients in each intervention group (ORNU, LRNU, and RRNU). Table 1 provides an overview of the clinicopathologic characteristics of the initial study population, the study population after multiple imputation, and the study population after propensity score matching. The PSM cohort’s median age was 72 years (IQR 64, 78), and 251 (33%) patients were female. Clinical and pathological demographics were similar among the three different surgical approaches, except for a significant difference between the years of surgery performed (*p* < 0.001), with an increasing use of LRNU from 2000–2005 and RRNU from 2006–2010. Among the perioperative characteristics, RRNU presented less intraoperative blood loss (100 mL (IQR 60, 200) vs. 150mL (IQR 75, 250) in LRNU vs. 200 mL (IQR 50, 350) in ORNU; *p* < 0.001), a shorter inpatient stay (4 days (IQR 2, 6) vs. 8 days (IQR 5, 14) after LRNU vs. 10 days (IQR 6, 15) after ORNU; *p* < 0.001), fewer overall postoperative complications (25.4% vs. 36.1% in LRNU vs. 39.1% in ORNU; *p* = 0.02), and fewer major postoperative complications (7.9% vs. 16.7% in LRNU vs. 20.2% in ORNU; *p* < 0.001).

### 3.2. Association of the Surgical Approach with Survival Outcomes 

During a median follow-up among the PSM cohort of 32 months (IQR 15, 61), 191 patients experienced disease recurrence, 292 developed bladder recurrence, 111 patients died of disease, and 86 patients died from other causes. 

The 3-year estimates for RFS, BRFS, CSS, and OS, as well as the associated Kaplan–Meier curves and pairwise log-rank tests, are provided in Figure 1A–D. While there was no difference in the pairwise log-rank tests for the 3-year RFS, CSS, and OS between the three surgical approaches, improved BRFS was observed for ORNU (3-year BRFS: 73.5% (67.7–79.8)) compared to LRNU (3-year BRFS: 58.8% (52.5–65.8); pairwise log-rank: *p* < 0.001) and RRNU (3-year BRFS: 58.9% (51.9–66.7); pairwise log-rank: *p* < 0.001).

Variables associated with RFS, BRFS, CSS, and OS are shown in Table 2. In the multivariable Cox regression analyses adjusted for possible confounders which were significantly associated with the according survival outcome in univariable regression analyses, both laparoscopic approaches (LRNU and RRNU) were independently associated with worse BRFS (hazard ratio (HR) 1.66, 95% confidence interval (CI) 1.22–2.28, *p* = 0.001 and HR 1.73, CI 1.22–2.47, *p* = 0.002, respectively). 

In the univariable and multivariable Cox regression analyses assessing the associations with RFS, CSS, and OS, the surgical approach was not shown to be associated with RFS, CSS, or OS in either the univariable or multivariable analyses.

### 3.3. Association of the Surgical Approach with Perioperative Outcomes 

Table 3 provides the associations of variables with perioperative outcomes. On multivariable regression analyses adjusted for possible confounders which were significantly associated with the according perioperative outcome in univariable regression analyses, the RRNU approach was independently associated with a shorter hospital stay (beta −6.1, CI −7.2–5.0; *p* < 0.001), fewer overall postoperative complications (odds ratio (OR) 0.53, CI 0.36–0.78; *p* = 0.001) and fewer MPCs (OR 0.27, CI 0.16–0.46; *p* < 0.001). Similarly, LRNU was independent associated with a shorter hospital stay (beta −1.1, CI −2.3–0.02; *p* = 0.047) and fewer MPCs (OR 0.50, CI 0.31–0.79; *p* = 0.003). There were no significant associations in the uni- and multivariable regression analyses between surgical approaches and intraoperative blood loss.

## 4. Discussion

In the present study, we investigated the association of the surgical RNU approach with the survival and perioperative outcomes in a large, propensity-score-matched cohort of patients suffering from non-metastatic UTUC treated with either ORNU, LRNU, or RRNU. We found no differences in terms of RFS, CSS, and OS between the three approaches. In contrast, we identified minimally invasive procedures (LRNU and RRNU) as independent predictors of worse BRFS. Moreover, we found an independent association between the RRNU approach with a shorter hospital length of stay and lower rates of overall and major complications. 

Several previous studies investigated the impact of the surgical RNU approach on survival outcomes [6,8,13,16,17,26,27,28]. Most of these retrospective studies reported similar outcomes in terms of progression-free survival (PFS), CSS, and OS between the three different surgical approaches despite their different statistical approaches, methods, and cohort sizes [8,13,26,27,28]. The most recent meta-analysis by Vecchia et al., which investigated the effect of the surgical technique (ORNU, LRNU, or RRNU) on PFS and CSS in approximately 87,000 patients, observed no significant difference [16]. Thus far, the only prospective study in the current literature that has investigated whether ORNU or LRNU has an impact on survival outcomes reported no differences in PFS and CSS between the two procedures [6]. Overall, these homogeneous results from different studies are consistent with the findings in our study, and thus strengthen the previous evidence that the choice of surgical method does not seem to affect RFS/PFS, CSS, and OS. However, it should be noted that these survival measures contain mainly estimated 5-year survival data, and there is only little evidence on long-term survival outcomes after LRNU and RRNU.

In the present study, we found a significant and independent association between LRNU and RRNU with a worse BRFS compared to ORNU. According to the current literature, BRFS as an endpoint related to the different surgical approaches has been investigated in only a few studies thus far, and the results are conflicting [6,26,27]. Simone et al., who conducted the first and only small-scale prospective study comparing survival outcomes between LRNU and ORNU, found no differences in BRFS between the two approaches [6]. Similar results were reported by Clements et al. in 2018, who found no difference among ORNU, LRNU, or RRNU in 3801 UTUC patients [26]. In contrast, in a propensity-score-matched cohort of 1276 patients, Kim et al. 2019 demonstrated that LRNU has a favorable effect on BRFS [27].

In the current literature, there are studies that identified predictors of worse BRFS and whose results partially support our findings [29,30,31]. Xylinas et al. showed, in an extensive series of North American UTUC patients, that the laparoscopic surgical technique and endoscopic management of the distal ureter were independent predictors of worse BRFS, which was successfully validated in a European cohort [29,30]. Shigeta et al. found that LRNU with a pneumoperitoneum time >150min was a strong, independent predictor of worse BRFS [31]. In a series of ORNU, it was demonstrated that distal ureteral clipping prior to mobilization of the kidney and ureter has a positive impact on BRFS [32]. Pizzighella et al. identified distal ureteral tumor localization as a risk factor for worse BRFS [33]. Furthermore, BCE techniques used with open or laparoscopic approaches and their impact on BRFS have been investigated in a few small studies, with heterogenous results being obtained thus far [33,34,35]. Therefore, the causes of bladder recurrence after RNU might be manifold, including not only the surgical approach but also the influence of the surgical technique with regard to distal ureteral management, surgery duration, and tumor location. Whether the surgeon’s experience and dexterity, which may play a role in laparoscopic approaches, has an influence, has not been yet investigated. The fact that only about one-eighth of patients in the present study received perioperative intravesical chemotherapy further supports the previous recommendation that such a therapy should be performed, as it may improve BRFS, especially for minimally invasive approaches [36]. 

In direct comparison, our PSM cohort showed increasingly better and significantly different perioperative parameters from ORNU to LRNU and to RRNU. Blood loss, the number and severity of postoperative complications, and the hospital length of stay were all better in the RRNU group, followed by LNRU, and worst after ORNU. In the multivariable regression analysis, we found a significant benefit for RRNU and LRNU compared to ORNU. Our findings regarding complications and hospital length of stay are consistent with previous larger-scale studies which performed a similar robust statistical analysis [4,37]. However, the three surgical techniques had few major complications and, given the similar survival outcomes, all approaches can be considered as feasible treatment options for patients suffering from non-metastatic UTUC.

Our study is not free of limitations. First, its retrospective and multicenter nature is characterized by the lack of strictly standardized protocols for patient selection and postoperative follow-up, which implies selection bias. Moreover, it should be noted that different surgeons with different surgical techniques and experiences may influence the present results. For example, we only had precise information on the BCE technique and if a ureter clipping was performed in a minority of patients; therefore, this could not have been augmented by multiple imputation. Furthermore, the retrospective setting means that patients are usually followed up by general practitioners after an average follow-up period of 2–3 years, which resulted in shorter survival data available to us. Future studies are required to assess the impact of BRFS on long-term survival as well on patient’s quality of life. Despite the fact that the present data come from several developed countries from all over the world with well-equipped and medically trained centers, it must be noted that the results of this study probably cannot be generalized to developing countries.

However, this is the first large-scale study comparing survival and perioperative outcomes between ORNU, LRNU, and RRNU, whereby detailed information on patient, preoperative, perioperative, postoperative, surgical, and pathological characteristics are available. Moreover, we were able to impute missing variables using a statistically powerful method which uses prediction models to create multiple data sets, and thus could impute meaningful values compared to other imputation methods, reducing the sizes of the standard errors at the same time.

## 5. Conclusions

In the present study, we confirmed previous findings that ORNU, LRNU, and RRNU have comparable RFS, CSS, and OS. Moreover, minimal invasive procedures (LRNU and RRNU) were shown to have a significantly worse BRFS. In order to identify the associated factors that promote bladder recurrence, additional studies are warranted to further improve the minimal invasive techniques in the future. In terms of perioperative outcomes, LRNU and RRNU were independently associated with shorter hospital length of stay and fewer major complications. However, all surgical procedures appear to be generally safe treatment options.

## Figures and Tables

**Figure 1 cancers-15-01409-f001:**
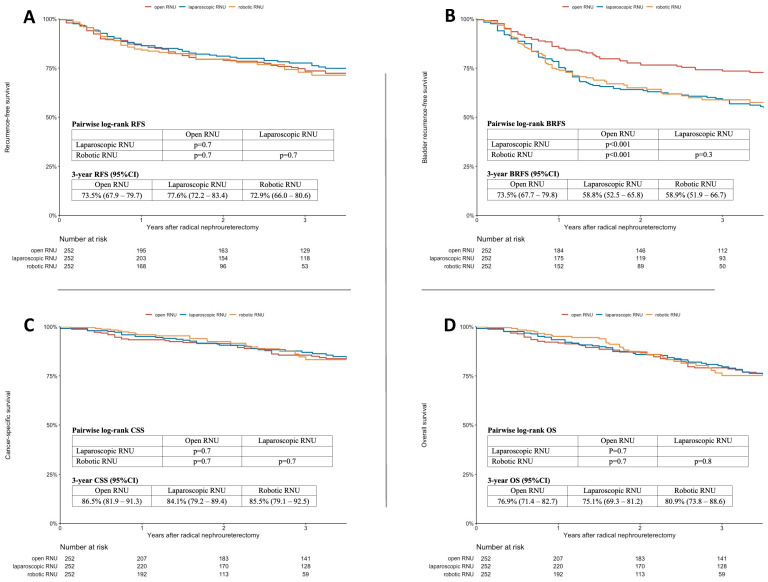
Kaplan–Meier analysis and log-rank tests for recurrence-free survival (**A**), bladder recurrence-free survival (**B**), cancer-specific survival (**C**), and overall survival (**D**) of a multiple-imputed and propensity-score-matched cohort of 756 patients treated with radical nephroureterectomy for UTUC stratified by the surgical approach.

**Table 1 cancers-15-01409-t001:** Baseline, peri-, and postoperative characteristics of patients with UTUC stratified by the type of surgical RNU approach in the original cohort, in the cohort after multiple imputation, and in the cohort after multiple imputation and propensity score matching.

	Original Cohort					Multiple-Imputed Cohort					Multiple-Imputed and Propensity-Score-Matched Cohort				
	Total Cohort	Surgical Approach				Total Cohort	Surgical Approach				Total Cohort	Surgical Approach			
Characteristic	N = 2434	Open RNU, N = 1096	Laparoscopic RNU, N = 865	Robotic RNU, N = 473	*p*-Value	N = 2434	Open RNU, N = 1096	Laparoscopic RNU, N = 865	Robotic RNU, N = 473	*p*-Value	N = 756	Open RNU, N = 252	Laparoscopic RNU, N = 252	Robotic RNU, N = 252	*p*-Value
Age	71 (64, 77)	70 (63, 77)	72 (65, 78)	71 (63, 77)	<0.001	71 (64, 77)	70 (63, 77)	72 (65, 78)	71 (63, 77)	<0.001	72 (64, 78)	73 (66, 78)	72 (65, 78)	70 (62, 77)	0.05
Missing	4	3	1	0											
Female Gender	780 (32%)	326 (30%)	264 (31%)	190 (40%)	<0.001	781 (32%)	327 (30%)	264 (31%)	190 (40%)	<0.001	251 (33%)	85 (34%)	82 (33%)	84 (33%)	>0.9
Missing	3	3	0	0											
ASA					<0.001					<0.001					0.5
1	129 (6.8%)	30 (4.0%)	62 (9.0%)	37 (8.0%)		172 (7.1%)	50 (4.6%)	85 (9.8%)	37 (7.8%)		45 (6.0%)	10 (4.0%)	14 (5.6%)	21 (8.3%)	
2	863 (46%)	372 (50%)	292 (42%)	199 (43%)		1081 (44%)	534 (49%)	341 (39%)	206 (44%)		339 (45%)	116 (46%)	115 (46%)	108 (43%)	
3	855 (45%)	323 (44%)	325 (47%)	207 (45%)		1107 (45%)	480 (44%)	418 (48%)	209 (44%)		346 (46%)	115 (46%)	115 (46%)	116 (46%)	
4	48 (2.5%)	17 (2.3%)	10 (1.5%)	21 (4.5%)		74 (3.0%)	32 (2.9%)	21 (2.4%)	21 (4.4%)		26 (3.4%)	11 (4.4%)	8 (3.2%)	7 (2.8%)	
Missing	539	354	176	9											
BMI	25.8 (23.0, 28.9)	25.4 (22.6, 28.7)	25.7 (23.0, 28.6)	26.2 (23.8, 29.6)	<0.001	25.8 (23.0, 29.0)	25.6 (22.5, 29.0)	25.7 (22.9, 28.8)	26.2 (23.8, 29.6)	0.001	26 (23, 29)	25.6 (22.3, 28.9)	26.0 (23.0, 29.0)	26.0 (23.4, 29.0)	0.3
Missing	585	377	201	7											
Previous bladder cancer	649 (29%)	284 (32%)	253 (30%)	112 (24%)	0.009	720 (30%)	342 (31%)	266 (31%)	112 (24%)	0.007	216 (29%)	75 (30%)	77 (31%)	64 (25%)	0.4
Missing	233	200	29	4											
Hydronephrosis	846 (43%)	336 (42%)	300 (43%)	210 (45%)	0.5	1044 (43%)	447 (41%)	384 (44%)	213 (45%)	0.2	316 (42%)	91 (36%)	111 (44%)	114 (45%)	0.08
Missing	462	290	163	9											
Tumor location					<0.001					<0.001					0.5
Pelvicaliceal	1360 (64%)	617 (67%)	451 (62%)	292 (63%)		1611 (66%)	757 (69%)	558 (65%)	296 (63%)		487 (64%)	160 (63%)	163 (65%)	164 (65%)	
Ureter	683 (32%)	281 (30%)	257 (36%)	145 (31%)		746 (31%)	308 (28%)	290 (34%)	148 (31%)		244 (32%)	80 (32%)	81 (32%)	83 (33%)	
Both	66 (3.1%)	25 (2.7%)	14 (1.9%)	27 (5.8%)		77 (3.2%)	31 (2.8%)	17 (2.0%)	29 (6.1%)		25 (3.3%)	12 (4.8%)	8 (3.2%)	5 (2.0%)	
Missing	325	173	143	9											
Diagnostic ureteroscopy performed	1376 (67%)	570 (70%)	528 (66%)	278 (66%)	0.1	1656 (68%)	602 (70%)	743 (68%)	311 (66%)	0.3	507 (67%)	172 (68%)	165 (65%)	170 (67%)	0.8
Missing	391	52	290	49											
Neoadjuvant chemotherapy	122 (5.0%)	77 (7.0%)	27 (3.1%)	18 (3.8%)	<0.001	122 (5.0%)	77 (7.0%)	27 (3.1%)	18 (3.8%)	<0.001	18 (2.4%)	6 (2.4%)	6 (2.4%)	6 (2.4%)	>0.9
Missing	1	1	0	0											
Year of surgery					<0.001					<0.001					<0.001
1990–1999	129 (5.7%)	128 (14%)	1 (0.1%)	0		146 (6.0%)	145 (13%)	1 (0.1%)	0		27 (3.6%)	27 (11%)	0 (0%)	0 (0%)	
2000–2005	222 (9.8%)	127 (13%)	95 (11%)	0		238 (9.8%)	137 (12%)	101 (12%)	0		49 (6.5%)	22 (8.7%)	27 (11%)	0 (0%)	
2006–2010	424 (19%)	232 (25%)	184 (22%)	8 (1.7%)		480 (20%)	285 (26%)	187 (22%)	8 (1.7%)		129 (17%)	73 (29%)	50 (20%)	6 (2.4%)	
2011–2015	744 (33%)	277 (29%)	295 (35%)	172 (36%)		789 (32%)	320 (29%)	297 (34%)	172 (36%)		264 (35%)	81 (32%)	90 (36%)	93 (37%)	
2016–2020	737 (33%)	179 (19%)	265 (32%)	293 (62%)		781 (32%)	209 (19%)	279 (32%)	293 (62%)		287 (38%)	49 (19%)	85 (34%)	153 (61%)	
Missing	178	153	25	0											
Side of surgery					0.2					0.3					0.2
Left	1105 (50%)	483 (51%)	414 (49%)	208 (46%)		1199 (49%)	556 (51%)	423 (49%)	220 (47%)		367 (49%)	123 (49%)	132 (52%)	112 (44%)	
Right	1125 (50%)	459 (49%)	426 (51%)	240 (54%)		1235 (51%)	540 (49%)	442 (51%)	253 (53%)		389 (51%)	129 (51%)	120 (48%)	140 (56%)	
Missing	204	154	25	25											
Lymphadenectomy performed	851 (37%)	412 (41%)	245 (30%)	194 (43%)	<0.001	903 (37%)	444 (41%)	256 (30%)	203 (43%)	<0.001	260 (34%)	86 (34%)	78 (31%)	96 (38%)	0.2
Missing	154	86	47	21											
Blood loss (mL)	200 (100, 350)	300 (200, 500)	192 (100, 300)	100 (70, 200)	<0.001	200 (90, 250)	200 (100, 400)	150 (75, 250)	100 (60, 200)	<0.001	150 (75, 250)	200 (150, 350)	150 (751, 250)	100 (60, 200)	<0.001
Missing	895	498	361	36											
Surgery duration (min)	250.00 (190.25, 324.00)	245 (187, 320)	264 (210, 343)	240 (180, 310)	<0.001	200 (196, 312)	200 (196, 305)	200 (196, 319)	240 (181, 312)	0.006	201 (196, 310)	200 (196, 305)	208 (196, 319)	240 (182, 308)	0.4
Missing	1232	666	487	79											
Perioperative intravesical chemotherapy Instillation	240 (14%)	73 (11%)	112 (20%)	55 (13%)	<0.001	394 (16%)	149 (14%)	183 (21%)	62 (13%)	<0.001	100 (13%)	32 (13%)	30 (12%)	38 (15%)	0.5
Missing	762	408	306	48											
Pathological tumor stage					0.007					0.007					0.8
≤pT1	1251 (51%)	525 (48%)	484 (56%)	242 (51%)		1251 (51%)	525 (48%)	484 (56%)	242 (51%)		411 (54%)	134 (53%)	144 (57%)	133 (53%)	
pT2	365 (15%)	172 (16%)	114 (13%)	79 (17%)		365 (15%)	172 (16%)	114 (13%)	79 (17%)		109 (14%)	39 (15%)	32 (13%)	38 (15%)	
pT3/pT4	818 (34%)	399 (36%)	267 (31%)	152 (32%)		818 (34%)	399 (36%)	267 (31%)	152 (32%)		236 (31%)	79 (31%)	76 (30%)	81 (32%)	
Pathological tumor grade					0.02					0.06					0.6
Low grade	571 (24%)	243 (23%)	228 (28%)	100 (21%)		599 (25%)	259 (24%)	236 (27%)	104 (22%)		187 (25%)	57 (23%)	64 (25%)	66 (26%)	
High grade	1784 (76%)	817 (77%)	600 (72%)	367 (79%)		1835 (75%)	837 (76%)	629 (73%)	369 (78%)		569 (75%)	195 (77%)	188 (75%)	186 (74%)	
Missing	79	36	37	6											
Tumor multifocality	603 (26%)	295 (28%)	207 (24%)	101 (23%)	0.06	624 (26%)	304 (28%)	211 (24%)	109 (23%)	0.09	177 (23%)	57 (23%)	61 (24%)	59 (23%)	>0.9
Missing	82	42	10	30											
Number of lymph nodes removed	6.00 (3.00, 13.00)	6 (3, 12)	6 (2, 12)	8 (3, 14)	0.047	6 (3, 13)	6 (3, 12)	6 (2, 12)	8 (3, 14)	0.047	7 (3, 12)	6 (4, 11)	7 (3, 12)	7 (2, 14)	0.7
Missing	170	88	62	20											
Lymph node involvement	264 (11%)	147 (13%)	68 (7.9%)	49 (10%)	<0.001	264 (11%)	147 (13%)	68 (7.9%)	49 (10%)	<0.001	59 (7.8%)	16 (6.3%)	21 (8.3%)	22 (8.7%)	0.8
Lymphovascular invasion	399 (16%)	177 (16%)	136 (16%)	86 (18%)	0.5	400 (16%)	178 (16%)	136 (16%)	86 (18%)	0.5	124 (16%)	41 (16%)	38 (15%)	45 (18%)	0.7
Missing	1	1	0	0											
Concomitant carcinoma in situ	388 (16%)	170 (16%)	141 (16%)	77 (16%)	0.9	388 (16%)	170 (16%)	141 (16%)	77 (16%)	0.9	116 (15%)	37 (15%)	40 (16%)	39 (15%)	>0.9
Positive soft tissue surgical margins	75 (3.9%)	37 (5.1%)	24 (3.4%)	14 (3.0%)	0.1	109 (4.5%)	66 (6.0%)	29 (3.4%)	14 (3.0%)	0.004	31 (4.1%)	16 (6.3%)	9 (3.6%)	6 (2.4%)	0.07
Missing	534	368	163	3											
Variant histology	110 (5.1%)	38 (4.2%)	30 (3.7%)	42 (9.5%)	<0.001	135 (5.5%)	54 (4.9%)	35 (4.0%)	46 (9.7%)	<0.001	39 (5.2%)	13 (5.2%)	12 (4.8%)	14 (5.6%)	>0.9
Missing	267	186	50	31											
Highest complication (Clavien–Dindo classification)					0.1					<0.001					0.02
No complications	1345 (73%)	498 (72%)	510 (72%)	337 (78%)		1633 (67%)	703 (64%)	570 (66%)	360 (76%)		504 (67%)	153 (61%)	162 (64%)	189 (75%)	
1	112 (6.1%)	38 (5.5%)	55 (7.8%)	19 (4.4%)		132 (5.4%)	50 (4.6%)	63 (7.3%)	19 (4.0%)		39 (5.2%)	12 (4.8%)	16 (6.3%)	11 (4.4%)	
2	247 (13%)	98 (14%)	95 (13%)	54 (12%)		311 (13%)	139 (13%)	111 (13%)	61 (13%)		100 (13%)	36 (14%)	32 (13%)	32 (13%)	
3a	25 (1.4%)	11 (1.6%)	7 (1.0%)	7 (1.6%)		70 (2.9%)	28 (2.6%)	34 (3.9%)	8 (1.7%)		25 (3.3%)	8 (3.2%)	10 (4.0%)	7 (2.8%)	
3b	50 (2.7%)	19 (2.7%)	21 (3.0%)	10 (2.3%)		115 (4.7%)	62 (5.7%)	40 (4.6%)	13 (2.7%)		35 (4.6%)	13 (5.2%)	14 (5.6%)	8 (3.2%)	
4a	33 (1.8%)	21 (3.0%)	9 (1.3%)	3 (0.7%)		109 (4.5%)	75 (6.8%)	27 (3.1%)	7 (1.5%)		32 (4.2%)	17 (6.7%)	11 (4.4%)	4 (1.6%)	
4b	7 (0.4%)	2 (0.3%)	4 (0.6%)	1 (0.2%)		36 (1.5%)	23 (2.1%)	11 (1.3%)	2 (0.4%)		12 (1.6%)	8 (3.2%)	4 (1.6%)	0 (0%)	
5	17 (0.9%)	8 (1.2%)	7 (1.0%)	2 (0.5%)		28 (1.2%)	16 (1.5%)	9 (1.0%)	3 (0.6%)		9 (1.2%)	5 (2%)	3 (1.2%)	1 (0.4%)	
Missing	598	401	157	40											
Number of patients with major complications (Clavien–Dindo classification ≥3)	132 (7.0%)	61 (8.5%)	48 (6.6%)	23 (5.1%)	0.08	358 (14.7%)	204 (18.6%)	121 (14%)	33 (7%)	<0.001	113 (44.8%)	51 (20.2%)	42 (16.7%)	20 (7.9%)	<0.001
Missing	540	375	143	22											
Hospital length of stay (days)	7 (4, 10)	9 (6, 12)	7 (4, 12)	4 (3, 5)	<0.001	8 (4, 13)	10 (7, 14)	7 (4, 13)	4 (3, 6)	<0.001	7 (4, 13)	10 (6, 15)	8 (5, 14)	4 (2, 6)	<0.001
Missing	789	489	272	28											
Adjuvant chemotherapy	346 (14%)	160 (15%)	127 (15%)	59 (12%)	0.5	346 (14%)	160 (15%)	127 (15%)	59 (12%)	0.5	100 (13%)	34 (13%)	37 (15%)	29 (12%)	0.6
Missing	3	3	0	0											
Adjuvant RT	44 (2.4%)	24 (3.2%)	13 (1.8%)	7 (2.0%)	0.2	88 (3.6%)	44 (4.0%)	21 (2.4%)	23 (4.9%)	0.05	34 (4.5%)	12 (4.8%)	6 (2.4%)	16 (6.3%)	0.1
Missing	588	341	123	124											

Median (IQR); n (%), Kruskal–Wallis rank-sum test; Pearson’s chi-squared test; Fisher’s exact test.

**Table 2 cancers-15-01409-t002:** Uni- and multivariable Cox regression analyses assessing the association of the three different surgical approaches with recurrence-free survival, bladder-recurrence-free survival, cancer-specific survival, and overall survival in a multiple-imputed and propensity-score-matched cohort of 756 patients treated with radical nephroureterectomy for UTUC.

	Recurrence-Free Survival						Bladder-Recurrence-Free Survival						Cancer-Specific Survival						Overall Survival					
	Univariable			Multivariable			Univariable			Multivariable			Univariable			Multivariable			Univariable			Multivariable		
Characteristic	HR	95% CI	*p*-Value	HR	95% CI	*p*-Value	HR	95% CI	*p*-Value	HR	95% CI	*p*-Value	HR	95% CI	*p*-Value	HR	95% CI	*p*-Value	HR	95% CI	*p*-Value	HR	95% CI	*p*-Value
Surgical approach (reference: open RNU)																								
Laparoscopic RNU	0.93	0.67, 1.31	0.7	0.98	0.70, 1.39	>0.9	1.81	1.35, 2.42	<0.001	1.66	1.22, 2.28	0.001	0.91	0.60, 1.39	0.7	0.92	0.59, 1.41	0.7	0.88	0.64, 1.21	0.4	0.93	0.68, 1.28	0.6
Robotic RNU	1.07	0.75, 1.53	0.7	1.03	0.71, 1.49	0.9	2.05	1.51, 2.77	<0.001	1.73	1.22, 2.47	0.002	0.79	0.48, 1.31	0.4	0.65	0.39, 1.10	0.1	0.83	0.56, 1.22	0.3	0.81	0.55, 1.19	0.3
Age	1.01	1.00, 1.03	0.1				1.00	0.99, 1.01	0.7				1.01	0.99, 1.03	0.2				1.03	1.01, 1.05	<0.001	1.03	1.01, 1.04	0.002
ASA (reference: 1)																								
2	1.00	0.52, 1.94	>0.9				1.52	0.84, 2.74	0.2	1.54	0.85, 2.80	0.2	0.64	0.32, 1.32	0.2				0.69	0.39, 1.22	0.2			
3	1.30	0.68, 2.50	0.4				1.50	0.83, 2.72	0.2	1.36	0.74, 2.50	0.3	0.86	0.42, 1.75	0.7				1.07	0.61, 1.88	0.8			
4	1.73	0.70, 4.26	0.2				2.70	1.26, 5.77	0.01	1.88	0.84, 4.20	0.1	1.00	0.31, 3.26	>0.9				1.01	0.39, 2.63	>0.9			
BMI	0.99	0.96, 1.01	0.4				1.03	1.01, 1.05	0.006	1.02	0.99, 1.04	0.1	0.97	0.94, 1.01	0.2				0.98	0.95, 1.01	0.2			
Gender (reference: male)																								
Female	1.08	0.80, 1.46	0.6				0.85	0.66, 1.09	0.2				1.00	0.68, 1.49	>0.9				1.03	0.77, 1.39	0.8			
Previous bladder cancer	1.18	0.87, 1.61	0.3				1.16	0.90, 1.49	0.2				1.21	0.81, 1.81	0.4				1.40	1.04, 1.88	0.03	1.56	1.15, 2.12	0.004
Hydronephrosis	1.22	0.92, 1.63	0.2				1.08	0.86, 1.37	0.5				1.44	0.99, 2.10	0.06				1.15	0.87, 1.54	0.3			
Tumor location (reference: pelvicaliceal)																								
Ureter	1.03	0.76, 1.40	0.9				0.86	0.67, 1.10	0.2				1.02	0.68, 1.52	>0.9				0.82	0.60, 1.12	0.2			
Both	0.71	0.31, 1.62	0.4				0.48	0.23, 1.02	0.06				1.00	0.40, 2.47	>0.9				0.65	0.30, 1.39	0.3			
Diagnostic ureteroscopy performed	1.05	0.77, 1.42	0.8				1.34	1.03, 1.74	0.03	1.35	1.03, 1.78	0.03	1.02	0.68, 1.52	>0.9				0.99	0.73, 1.34	>0.9			
Neoadjuvant chemotherapy	3.83	2.01, 7.27	<0.001	3.14	1.56, 6.29	0.001	0.64	0.24, 1.72	0.4				3.47	1.41, 8.57	0.007	2.28	0.89, 5.85	0.09	2.64	1.16, 5.97	0.02	1.62	0.69, 3.79	0.3
Year of surgery (reference: 1990–1999)																								
2000–2005	1.40	0.62, 3.14	0.4				1.54	0.72, 3.27	0.3	0.95	0.43, 2.10	0.9	1.54	0.60, 3.93	0.4				1.10	0.58, 2.09	0.8			
2006–2010	1.12	0.53, 2.35	0.8				1.13	0.57, 2.24	0.7	0.86	0.42, 1.75	0.7	0.85	0.35, 2.09	0.7				0.66	0.36, 1.20	0.2			
2011–2015	1.20	0.58, 2.48	0.6				1.31	0.68, 2.55	0.4	0.90	0.44, 1.84	0.8	0.86	0.36, 2.05	0.7				0.76	0.43, 1.36	0.4			
2016–2020	1.41	0.67, 2.96	0.4				2.82	1.45, 5.48	0.002	1.72	0.83, 3.59	0.2	1.06	0.43, 2.66	0.9				1.12	0.60, 2.09	0.7			
Perioperative intravesical chemotherapy instillation	1.17	0.77, 1.77	0.5				1.53	1.12, 2.10	0.008	1.51	1.08, 2.10	0.02	0.65	0.31, 1.33	0.2				0.80	0.48, 1.31	0.4			
Lymphadenectomy performed	1.32	0.98, 1.78	0.07				1.29	0.96, 1.74	0.09				1.66	1.13, 2.42	0.009	1.87	0.73, 4.79	0.2	1.27	0.94, 1.72	0.1			
Pathological tumor stage (reference: ≤pT1)																								
pT2	1.71	1.07, 2.74	0.03	1.48	0.91, 2.42	0.1	0.84	0.59, 1.19	0.3	0.89	0.62, 1.28	0.5	1.63	0.90, 2.94	0.1	1.20	0.65, 2.23	0.6	1.60	1.06, 2.43	0.03	1.23	0.80, 1.90	0.3
pT3/4	3.74	2.72, 5.13	<0.001	2.09	1.44, 3.03	<0.001	0.68	0.51, 0.90	0.006	0.71	0.53, 0.95	0.02	3.03	2.01, 4.55	<0.001	1.43	0.89, 2.31	0.1	2.11	1.55, 2.87	<0.001	1.21	0.85, 1.74	0.3
Pathological tumor grade (reference: low grade)																								
High grade	3.00	1.93, 4.65	<0.001	1.72	1.07, 2.75	0.03	0.92	0.71, 1.19	0.5				3.62	1.94, 6.75	<0.001	2.16	1.10, 4.24	0.03	2.43	1.63, 3.61	<0.001	1.71	1.12, 2.63	0.01
Variant histology	1.88	1.11, 3.19	0.02	0.93	0.54, 1.63	0.8	0.89	0.50, 1.59	0.7				1.31	0.57, 2.98	0.5				0.99	0.49, 2.01	>0.9			
Lymph node involvement	5.66	3.58, 8.93	<0.001	1.96	1.18, 3.27	0.009	0.44	0.22, 0.87	0.02	0.45	0.22, 0.90	0.02	8.30	4.63, 14.9	<0.001	3.48	1.83, 6.64	<0.001	6.00	3.69, 9.74	<0.001	3.31	1.96, 5.59	<0.001
Tumor multifocality	1.39	1.02, 1.91	0.04	1.31	0.94, 1.82	0.1	1.38	1.07, 1.78	0.02	1.32	1.01, 1.73	0.04	1.76	1.18, 2.61	0.005	1.61	1.06, 2.44	0.02	1.28	0.93, 1.76	0.1			
Number of lymph nodes removed	1.01	0.99, 1.03	0.4				1.00	0.99, 1.02	0.6				1.01	0.99, 1.04	0.2				1.01	0.99, 1.03	0.5			
Positive soft tissue surgical margins	3.04	1.82, 5.08	<0.001	2.01	1.15, 3.49	0.01	0.96	0.51, 1.80	0.9				2.80	1.42, 5.55	0.003	1.64	0.79, 3.39	0.2	1.73	0.91, 3.27	0.09			
Lymphovascular invasion	3.85	2.84, 5.21	<0.001	1.73	1.20, 2.50	0.003	0.79	0.55, 1.12	0.2				3.93	2.64, 5.85	<0.001	1.91	1.19, 3.07	0.007	2.64	1.90, 3.67	<0.001	1.60	1.09, 2.35	0.02
Concomitant carcinoma in situ	1.52	1.07, 2.16	0.02	1.14	0.78, 1.66	0.5	1.47	1.10, 1.97	0.008	1.46	1.08, 1.97	0.01	1.60	1.02, 2.51	0.04	1.04	0.64, 1.69	0.9	1.37	0.96, 1.96	0.09			
Adjuvant chemotherapy	4.52	3.32, 6.16	<0.001	2.33	1.62, 3.35	<0.001	1.02	0.71, 1.46	>0.9				4.42	2.97, 6.58	<0.001	2.10	1.30, 3.39	0.002	2.69	1.91, 3.78	<0.001	1.88	1.26, 2.81	0.002
Adjuvant radiotherapy	2.32	1.42, 3.77	<0.001	1.60	0.96, 2.67	0.07	0.43	0.21, 0.87	0.02	0.40	0.20, 0.82	0.01	1.57	0.77, 3.23	0.2				1.34	0.74, 2.40	0.3			

HR = hazard ratio, CI = confidence interval.

**Table 3 cancers-15-01409-t003:** Uni- and multivariable linear and logistic regression analyses predicting intraoperative blood loss, hospital length of stay, overall complications, and major complications in a multiple-imputed and propensity-score-matched cohort of 756 patients treated with radical nephroureterectomy for UTUC.

	Intraoperative Blood Loss						Hospital Length of Stay						Overall Postoperative Complications (Clavien–Dindo Classification 1–5 )						Major Postoperative Complications (Clavien–Dindo Classification ≥3)					
	Univariable			Multivariable			Univariable			Multivariable			Univariable			Multivariable			Univariable			Multivariable		
Characteristic	Beta	95% CI	*p*-Value	Beta	95% CI	*p*-Value	Beta	95% CI	*p*-Value	Beta	95% CI	*p*-Value	OR	95% CI	*p*-Value	OR	95% CI	*p*-Value	OR	95% CI	*p*-Value	OR	95% CI	*p*-Value
Surgical approach (reference: open RNU)																								
Laparoscopic RNU	95	−66, 256	0.2	95	−64, 254	0.2	−1.2	−2.3, −0.06	0.04	−1.1	−2.2, −0.02	0.047	0.86	0.60, 1.23	0.4	0.89	0.62, 1.28	0.5	0.50	0.32, 0.79	0.003	0.50	0.31, 0.79	0.003
Robotic RNU	−127	−289, 34	0.1	−140	−299, 20	0.09	−6.2	−7.3, −5.0	<0.001	−6.1	−7.2, −5.0	<0.001	0.52	0.35, 0.75	<0.001	0.53	0.36, 0.78	0.001	0.26	0.15, 0.44	<0.001	0.27	0.16, 0.46	<0.001
Age	6.1	−0.70, 13	0.08				0.06	0.01, 0.11	0.03	0.00	−0.05, 0.05	>0.9	1.01	0.99, 1.02	0.3				0.99	0.97, 1.01	0.4			
ASA (reference: 1)																								
2	−232	−519, 55	0.1				1.0	−1.1, 3.2	0.4				1.82	0.85, 3.91	0.1	1.79	0.82, 3.90	0.1	2.54	0.76, 8.49	0.1	2.07	0.61, 7.04	0.2
3	−44	−331, 243	0.8				−1.0	−3.1, 1.2	0.4				2.12	0.99, 4.56	0.05	2.10	0.97, 4.56	0.06	2.53	0.76, 8.47	0.1	2.13	0.63, 7.22	0.2
4	−216	−662, 230	0.3				1.8	−1.5, 5.2	0.3				7.56	2.54, 22.45	<0.001	7.51	2.48, 22.78	<0.001	10.27	2.52, 41.89	0.001	8.16	1.95, 34.24	0.004
BMI	−9.6	−22, 2.6	0.1				−0.23	−0.32, −0.14	<0.001	−0.20	−0.28, −0.11	<0.001	1.00	0.97, 1.03	>0.9				0.98	0.95, 1.02	0.4			
Gender (reference: male)																								
Female	192	52, 332	0.007	187	49, 326	0.008	0.65	−0.42, 1.7	0.2				0.71	0.51, 0.98	0.04	0.70	0.50, 0.98	0.04	0.85	0.56, 1.30	0.5			
Side (reference: left)																								
Right	−45	−178, 87	0.5				−0.53	−1.5, 0.48	0.3				0.83	0.62, 1.13	0.2				0.59	0.39, 0.87	0.01	0.60	0.40, 0.91	0.02
Neoadjuvant chemotherapy	−42	−475, 392	0.9				−4.6	−7.9, −1.3	0.006				0.39	0.11, 1.37	0.1				0.00	0.00, 0.00	>0.9			
Surgery duration	0.76	0.11, 1.4	0.02	0.78	0.13, 1.4	0.02	0.00	0.00, 0.01	0.09				1.00	1.00, 1.00	0.7				1.00	1.00, 1.00	0.2			
Lymphadenectomy performed	100	−39, 239	0.2				−0.05	−1.1, 1.0	>0.9				0.81	0.59, 1.12	0.2				0.80	0.52, 1.22	0.3			
Number of lymph nodes removed	2.6	−6.5, 12	0.6				−0.04	−0.11, 0.03	0.3				0.99	0.97, 1.01	0.5				0.98	0.95, 1.01	0.2			
Perioperative intravesical chemotherapy instillation	-	-	-				−1.9	−3.4, −0.46	0.01	−1.6	−3.0, −0.30	0.02	0.88	0.56, 1.39	0.6				0.78	0.42, 1.44	0.4			
Pathological tumor stage (reference: ≤pT1)																								
pT2	102	−93, 296	0.3	109	−84, 301	0.3	1.3	−0.20, 2.7	0.09	1.0	−0.32, 2.4	0.1	1.59	1.03, 2.46	0.04	1.77	1.13, 2.77	0.01	1.17	0.68, 2.04	0.6			
pT3/4	287	140, 435	<0.001	275	129, 421	<0.001	2.9	1.8, 4.0	<0.001	2.9	1.9, 4.0	<0.001	1.18	0.84, 1.66	0.3	1.20	0.84, 1.71	0.3	0.85	0.54, 1.34	0.5			

Beta = regression coefficient, OR = odds ratio, CI = confidence interval.

## Data Availability

The data presented in this study are available upon request from the corresponding author. The data are not publicly available due to privacy restrictions.

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
