# Peer review of "Comparing Oncological and Perioperative Outcomes of Open versus Laparoscopic versus Robotic Radical Nephroureterectomy for the Treatment of Upper Tract Urothelial Carcinoma: A Multicenter, Multinational, Propensity Score-Matched Analysis"

_cancers, 2023, doi:10.3390/cancers15051409_

Round 1
Reviewer 1 Report
This work is a retrospective study assessing the impact of surgical technique in postoperative outcomes and survival rates of patients undergoing nephroureterectomy for upper urinary tract tumors. The advantages of minimal invasiveness are more or less expected by the reader. The non-superiority of one technique over the other, regarding survival rates, has been also demosntrated by different research groups. The indriguing result is the worse bladder recurrence rates when applying minimally invasive techniques.
The study is well designed and presented, the language of the manuscript is acceptable for publication and the references are carefully selected. I commend the authors and recommend the acceptance fo the paper in its current form.
Author Response
We would like to thank the reviewer for his expressed appreciation and the critical review of our manuscript.
Reviewer 2 Report
This authors collected vast UTUC cases from different nations and different areas in US. This indeed lower the bias from biodiversity of UTUC. Generally, this paper provides fruitful, insightful, and sound results regarding the perioperative and oncological outcomes of UTUC. I would recommend the editor to publish this article due to so much merits of it. I have only some minor and personal interests questions:
1. I noted that the cases were mostly collected from the developed countries, but the authors mentioned in the Introduction section that UTUC seems more common in developing countries. Would that data bring any bias to the results ? such as, UTUC in developing countries might have worse perioperative and oncological outcomes due to the inferior medical equipment, delayed diagnosis, or less accessible to well-trained surgeons ? This way, this article might only reflect the optimal results of UTUC in real world ? Any literature discuss on this kind of topics ?
2. Did the author ever consider divide the tumors location into proximal, mid, and distal ureter ? cuz the authors mentioned that one of the reasons why open would be superior to the laparoscopic or robotic surgeries in bladder recurrence was in manipulating the distal ureter. Plus, since the pneumoperitoneal time was a strong independent indicator in predicting bladder recurrence and the HR of laparoscopy and robot seemed to be similar without difference when reference to open in this article, would the pneumoperitoneal time be alike in those two groups as well in this data ? Cuz, theoretically robot needs re-docking of arms and ports, while laparoscopy won't need it. So, the former would be around 20 minutes longer than the latter.
Author Response
This authors collected vast UTUC cases from different nations and different areas in US. This indeed lower the bias from biodiversity of UTUC. Generally, this paper provides fruitful, insightful, and sound results regarding the perioperative and oncological outcomes of UTUC. I would recommend the editor to publish this article due to so much merits of it. I have only some minor and personal interests questions:
- I noted that the cases were mostly collected from the developed countries, but the authors mentioned in the Introduction section that UTUC seems more common in developing countries. Would that data bring any bias to the results ? such as, UTUC in developing countries might have worse perioperative and oncological outcomes due to the inferior medical equipment, delayed diagnosis, or less accessible to well-trained surgeons ? This way, this article might only reflect the optimal results of UTUC in real world? Any literature discuss on this kind of topics ?
Answer: We thank the reviewer for this comment. There was a typo in the introduction and UTUC is more common in developed countries which we have revised accordingly. Moreover, we have included the latest cancer statistics (from 2021) and the corresponding reference.
Page 4: Urothelial cancers are the sixth most common tumors in developed countries, of which upper tract urothelial cancers (UTUC) account for only 5-10% (1)
Nevertheless, we agree with the reviewer that our results, which were obtained in well-equipped academic centers with trained surgeons, cannot be applied to developing countries. We have noted this accordingly in the limitations section. To the best of our knowledge, there are no studies on perioperative and oncologic outcomes in developing countries in the current literature.
Page 12: Despite that the present data comes from several developed countries from all over the world with well-equipped and medically trained centers, it must be noted that the results of this study probably cannot be generalized to developing countries.
- Did the author ever consider divide the tumors location into proximal, mid, and distal ureter ? cuz the authors mentioned that one of the reasons why open would be superior to the laparoscopic or robotic surgeries in bladder recurrence was in manipulating the distal ureter.
Answer: We thank the reviewer for these comments. Because tumor location was not associated with BRFS in our Cox regression, we decided not to perform further analysis on this matter.
Plus, since the pneumoperitoneal time was a strong independent indicator in predicting bladder recurrence and the HR of laparoscopy and robot seemed to be similar without difference when reference to open in this article, would the pneumoperitoneal time be alike in those two groups as well in this data ? Cuz, theoretically robot needs re-docking of arms and ports, while laparoscopy won't need it. So, the former would be around 20 minutes longer than the latter.
Answer: We agree with the reviewer that due to the additional docking in robotic surgery, a longer operation/pneumoperitoneum time can be expected. However, it should be noted that docking in robotic surgery takes a few minutes and port placement is the main time required which is also needed in laparoscopic surgery. Moreover, the laparoscopic approach using rigid instruments is a more challenging technique that results in a longer operation time. In addition, the experience of the surgeon must be considered. In addition, it should be noted that due to the retrospective multicenter design, there is no clear definition of the start and end of the procedure and therefore the exact operation times may vary minimally. We therefore believe that in the present study it is not possible to answer whether the pneumoperitoneal time of both groups is similar.
Reviewer 3 Report
The study aims to compare oncological and perioperative outcomes of open versus laparoscopic versus robotic radical nephroureterectomy for the treatment of upper tract urothelial carcinoma. Despite the retrospective design the wide sample size allows to reach solid results and conclusions
Author Response
We thank the reviewer for the positive feedback concerning our results and for the critical review of the manuscript.